

# Contraceptive use among reproductive-age females with disabilities in central Sidama National Regional State, Ethiopia: a multilevel analysis

Zelalem Tenaw[1], Taye Gari[2] and Achamyelesh Gebretsadik[2]

[1] Midwifery, Hawassa University, Hawassa, Sidama, Ethiopia
[2] Public Health, Hawassa University, Hawassa, Sidama, Ethiopia

## ABSTRACT

**Background.** Contraceptive use is an important and cost-effective intervention to prevent unwanted pregnancies. People with disabilities face discrimination when it comes to using contraception and are doubly burdened by unwanted pregnancies. However, the status of contraceptive use and associated factors among reproductive-aged females with disabilities was not adequately determined in Ethiopia.

**Objective.** This study aimed to assess contraceptive use and associated factors among reproductive-age females with disabilities in Dale and Wonsho districts and Yirgalem city administration of central Sidama National Regional State, Ethiopia.

**Methods.** A community-based cross-sectional study was conducted among randomly selected 620 reproductive-age females with disabilities living in the selected districts from June 20 to July 15, 2022. The data were collected through face-to-face interviewing techniques using a structured questionnaire. A multilevel logistic regression analysis model was employed to analyze the data. The adjusted odds ratio (AOR) with a 95% confidence interval (CI) was used to report the measures of associations.

**Results.** In this study, 27.3% (95% CI [23.8%–31.0%]) of the reproductive-age females with disabilities were current contraceptive users. Regarding the methods, 82 (48.5%) of the reproductive-age females with disabilities used implants. Having good contraceptive knowledge (AOR = 9.03; 95% CI [4.39–18.6]), transport accessibility to health facilities (AOR = 2.28; 95% CI [1.32–3.94]), being an adult (25 to 34 years old) (AOR = 3.04; 95% CI [1.53–6.04]), having a hearing disability (AOR = 0.38; 95% CI [0.18, 0.79]), having paralysis of the extremities (AOR = 0.06; 95% CI [0.03–0.12]), and wheel-chaired disability (AOR = 0.10; 95% CI [0.05–0.22]) were factors associated with contraceptive use.

**Conclusion.** Contraceptive use among reproductive-age females with disabilities is low. Transport accessibility, contraceptive knowledge, being in the age groups of 25 to 34 years, and the types of disability determine their contraceptive use. Therefore, designing appropriate strategies to provide contraceptive education and information and provide contraceptive services in their homes is important to enhance contraceptive use.

Corresponding author
Zelalem Tenaw,
abigiatenaw@gmail.com

## INTRODUCTION

More than one billion people in the world are estimated to have disabilities. The majority of them are from developing countries (*Zziwa et al., 2019*). People with disabilities are the most discriminated against and marginalized group in many countries, particularly in developing countries, including Ethiopia (*MacKay, 2006*; *Hosseinpoor et al., 2013*). In Sidama National Regional State, Ethiopia, we have observed that many females with disabilities are sacrificing themselves with unwanted pregnancies and unwanted children. Due to these problems, women with disabilities have been facing many challenges and have spent their lives begging around the roads and in churches with their unwanted pregnancies and children.

Contraceptive methods are chemicals, drugs, and surgical procedures used to prevent unwanted pregnancy (*Jain & Muralidhar, 2011*). Although people with disabilities have a reproductive right to access and use contraception, coverage of contraceptive use in developing countries, including Ethiopia, is low when compared to developed countries. This is evidenced by the fact that 70.1% (*Haynes et al., 2018*) of disabled women in the United States of America, 67.4% (*Aslan, Yılmaz & Acar, 2021*) in Turkey, 34% (*Olajide et al., 2014*) in Nigeria, 26.9% (*Trani et al., 2011*) in Sierra Leone, 26.1% (*Ayiga & Kigozi, 2016*) in Uganda, 17% (*Kumi-Kyereme, 2021*) in Ghana, 16% (*Odhiambo, 2012*) in Kenya, and 18% (*Beyene, Munea & Fekadu, 2019*) to 34% (*Yesgat et al., 2020*) in Ethiopia have access to contraceptives.

In Ethiopia, various factors associated with contraceptive use among females with disabilities were identified. Of the reported factors, marital status, age, types of disabilities, knowledge and attitude towards family planning methods, the presence of nearby health facilities providing family planning services, keeping confidentiality and privacy in the health facility, having a good self-perception, and educational and economic status were the most common (*Tsegay, Gebremariam & Haile, 2017*; *Beyene, Munea & Fekadu, 2019*; *Yimer Awol Seid, 2019*; *Gonie et al., 2020*; *Yesgat et al., 2020*). Not only do the factors listed above influence contraceptive use, but disability, such as types of disabilities and contraceptive methods, also has an administrative impact on contraceptive use.

In Ethiopia, few studies were conducted to determine the prevalence of contraceptive use and associated factors among reproductive-age females with disabilities from 2013 to 2019 (*Tsegay, Gebremariam & Haile, 2017*; *Beyene, Munea & Fekadu, 2019*; *Yimer Awol Seid, 2019*; *Gonie et al., 2020*; *Yesgat et al., 2020*). These studies considered only urban female residents, deaf and blind females, and females enrolled in supporting organizations and considered only individual-level factors. Contraceptive coverage in these populations is also inconsistent, ranging from 18% (*Beyene, Munea & Fekadu, 2019*) to 34% (*Yesgat et al., 2020*).

Therefore, this study aimed to determine the prevalence of contraceptive use and its associated factors among reproductive-age females with disabilities by considering rural and urban residency, all types of disability (except mental disability), and individual and community-level factors.

## METHODS AND MATERIALS

### Study design and setting

A community-based cross-sectional study was conducted from June 20 to July 15, 2022, to determine the prevalence and factors associated with contraceptive use among reproductive-age females with disabilities in Sidama National Regional State, Ethiopia. The study was conducted in the Dale and Wonsho districts and in the Yirgalem city administration. According to the *Sidama Region Health Bureau (2021)*, the total population of Dale and Wonsho districts and Yirgalem city administration was 469,455 (*Sidama Region Health Bureau, 2021*; *Sidama Region Health Bureau, 2022*). The two districts are the health and demographic surveillance sites of Hawassa University. Both districts are known for their coffee production and highly dense populations. In the districts and city administration, there are 56 rural and 10 urban kebeles (the lowest political administrative units in Ethiopia). The districts and city administration have one hospital, 16 health centers, and 54 health posts.

### Population

Reproductive-age females with disabilities in Dale and Wonsho districts and Yirgalem city administration in Sidama National Regional State were the source population. Reproductive-age females with disabilities who lived in the selected kebeles for at least six months were the study population, except for those who have dual disabilities (*i.e.,* cannot see and hear) and are seriously ill during the data collection time.

### Sample size and sampling procedure

The sample size for the first objective (prevalence) was determined by using Epi-Info version 7 software with the assumptions of a 95% confidence interval with 33.7% contraceptive use among reproductive-age women with disability (*Yesgat et al., 2020*), a level of significance ($\alpha$) of 0.05, a 5% margin of error ($d = 0.05$), and a design effect of 1.64. The sample size for factors associated with contraceptive use was also computed using Epi-Info version 7 with the assumptions of a two-sided confidence level of 95%, a power of 80, a ratio of one (unexposed: exposed), and a percent outcome in the unexposed group (14.5) *versus* a percent outcome in the exposed group (24.5). Accordingly, the maximum (530) sample size was determined by marital status (*Beyene, Munea & Fekadu, 2019*). The sample size from the prevalence of 563 was larger than the associated factors' maximum sample size of 530. After adjusting for an anticipated 10% nonresponse rate, the final sample size was 620.

The sample size was proportionally allocated to the randomly selected 30 kebeles (20 rural and 10 urban) based on the number of reproductive-age females with disabilities. Before conducting this study, a house-to-house census was done to determine the number and identify reproductive-age females with disabilities in each kebele. Reproductive-age females with disabilities were registered during the census using the tracing form. The registration form was used to select study participants using a simple random sampling technique.

## Variables

The outcome variable was contraceptive use. Whereas the independent variables were marital status, age, types of disability, educational status, knowledge about family planning, income, self-perception, attitude toward family planning, health care providers' attitudes, the presence of family planning provision at a nearby health facility, and the keeping of confidentiality and privacy by the health facility.

## Data collection procedures and quality assurance

The questionnaires (data collection tools) were developed by reviewing different existing literature, like EDHS 2016 (*Central Statistical Agency (CSA) [Ethiopia] and ICF, 2016*; *Gonie et al., 2020*; *Yesgat et al., 2020*), which consists of personal and socio-demographic characteristics and contraceptive use-related issues. After developing and pretesting the data collection tool, six data collectors and one supervisor who are fluent speakers of Sidamu Afoo and who have data collection experience were employed. The data were collected through face-to-face interviewing techniques using structured questionnaires. Two of the data collectors were proficient in sign language and collected the data from reproductive-age females with hearing disabilities. The interview was conducted in a place where confidentiality and privacy are assured. To assure the quality of the data collection, a three-day data collector training was given. The data collection tool was first prepared in English and then translated into Afoo-Sidamu, a local language, and then back to English to check the consistency. The trained data collectors did a pre-test on 31 (5%) reproductive-age females with disabilities in *Lokie kebele* Hawassa city to check the tools, and corrections were made based on the feedback. The principal investigator (PI) monitors and controls the overall process of data collection and makes appropriate corrections for any issues raised during data collection. The PI also checked the completeness of the questionnaires daily.

## Data management and analysis

The Kobo Collect version 2021.3.4 application was used to collect the data. Following collection, the data were imported into Stata version 16 for analysis using the "SSC install kobo2stata" command. The cleaning and organizing of the data were done in Stata. The types of variables were clarified, and the distribution was checked by running the frequency for categorical data and mean $\pm$ SD (standard deviation) for continuous variables. A multilevel logistic regression analysis model was used to account for the kebele level. Before using the multilevel logistic analysis model, we checked the intraclass correlation coefficient (ICC) level with the chi-square significance level to determine whether using the multilevel logistic analysis model is justifiable. The ICC of 0.12 and its chi-square ($P = 0.001$) significance level showed that using a multilevel analysis model is reasonable. Then, bi-variable multilevel logistic regression was done to identify eligible variables ($P$-value < 0.20) for multivariable multilevel logistic regression analysis. The multivariable multilevel logistic regression was performed to check for the presence of an association between level one or level two variables and contraceptive use. To determine whether a significant association existed and its strength, variables with adjusted odds ratios with a 95% confidence interval and a $P$-value < 0.05 were considered.

## Ethical considerations

The ethical clearance was obtained from the Institutional Review Board at the College of Medicine and Health Sciences of Hawassa University with an approval number of Ref. No. IRB/143/14. After approval, a support letter was written to the Sidama National Regional Public Health Institute. Then, after obtaining the support letter from Sidama National Regional Public Health Institute, the permission and cooperation letter were given to the woreda health offices. Finally, the woreda health offices wrote a permission letter to selected kebeles, asking them to cooperate and give consent to conduct the study. Written consent was obtained from the study participants to collect the data. There is no risk in participating in this survey. People with disabilities having different health problems were linked to nearby health facilities for possible support and follow-up.

# RESULTS

## Socio-demographic characteristics of study participants

A total of 620 reproductive-age females with disabilities were included in this study. The mean (SD) age of the study participants was 28.12 (8.54) years. Of the study participants, 55.32% had no formal education (illiterate) and almost all (98.90%) were not employed. Most (83.71%) of the reproductive-age females with disabilities had no occupation, and 54.20% were married (Table 1).

## Contraceptive knowledge and attitude

Among the study participants, 382 (61.6%) had good knowledge about contraceptives. Regarding attitude, 303 (48.9%) of reproductive-age females with disabilities had a positive attitude towards contraceptive use.

## Contraceptive use prevalence

In this study, the overall prevalence of current contraceptive use among reproductive-age females with disabilities was 27.3% (95% CI [23.8–31.0]), of which 19.19% (95% CI [16.17–22.52]) were from rural residents and 8.06 (95% CI [6.04–10.49]) were from urban residents. From the overall contraceptive use, 20.3% (95% CI [17.2–23.7]) were married, and 7% (95% CI [5.06–9.22]) were unmarried.

## Types of contraceptive methods used

Of the contraceptive method users, 82 (48.5%) of the reproductive-age females with disabilities used implants, followed by injectable (36%), oral contraceptive pills (12%), and intrauterine contraceptive devices and condoms (4%).

## Reasons for not using contraceptives and their plan to use in the future

This study tried to identify the possible reasons for not using contraceptives among the 451 non-users of contraceptives of reproductive age females with disabilities. Of the respondents, the majority, 161 (36%) did not use it due to a lack of information about contraceptives, and 23 (5%) are due to the plan to have a baby in the near future (Fig. 1).
**Table 1 Socio-demographic characteristics of reproductive-age females with disabilities in Sidama Regional Stata, Ethiopia, 2022 (N = 620).**

| Variable | | Number | Percent |
|---|---|---|---|
| Age in years mean (SD) | 28.12 (8.54) | | |
| Marital status of the participants | Married | 336 | 54.2 |
| | Never married | 239 | 38.6 |
| | Widowed | 20 | 3.20 |
| | Divorced | 25 | 4.00 |
| Residency | Rural | 381 | 61.45 |
| | Urban | 239 | 38.55 |
| Participants educational status | Illiterate | 343 | 55.32 |
| | Primary school | 192 | 30.96 |
| | Secondary school | 78 | 12.60 |
| | Vocational and technique | 7 | 1.12 |
| Employment status | Employed | 6 | 1.10 |
| | Unemployed | 538 | 98.90 |
| Wealth index of household | Low | 216 | 34.84 |
| | Medium | 196 | 31.61 |
| | High | 208 | 33.55 |

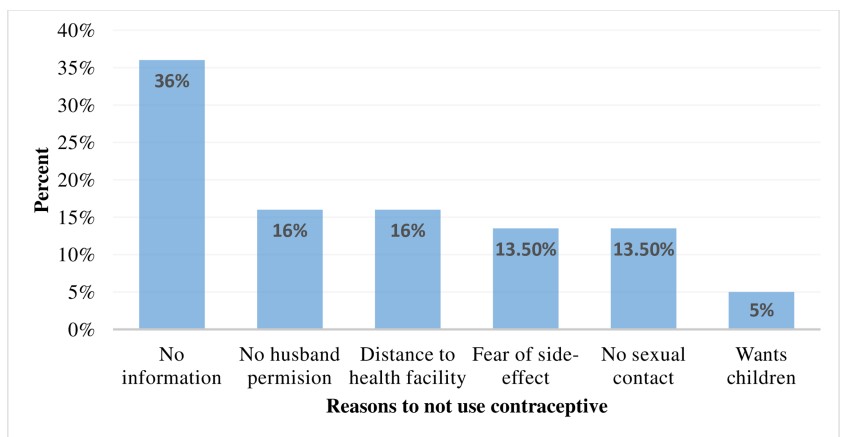

**Figure 1 Reasons for not using contraceptives among reproductive-age females with disabilities in Sidama National Regional State, Ethiopia, 2022.**

Regarding their future plans for contraceptive use, 147 (32.59%) had the plan to use, 209 (46.34%) had no plan to use, and 95 (21.06%) were not sure about their future plans.

## Factors associated with contraceptive use random effect model

In the zero model (model I), 12% of the variability in contraceptive use was at the community level (kebele level). This may be attributable to other unobserved community factors (ICC = 0.12), which were supported by the chi-square test ($P < 0.001$). This finding also showed that using a multilevel analysis model is reasonable.

## Fixed effect model

In the bivariable logistic regression, marital status, education, occupation, self-perception, age, transport accessibility, contraceptive knowledge, types of disability, and residence were significantly associated with contraceptive use, but in the multivariable multilevel logistic regression analysis (after adjusting for the possible confounders), contraceptive knowledge, transport accessibility to the health facility, age, and types of disability were significantly associated with contraceptive use.

Reproductive-age females with a disability who knew about contraceptives had nine (AOR = 9.03; 95% CI [4.39–18.6]) times higher odds of contraceptive use compared with those who had no contraceptive knowledge. On the other hand, reproductive-age females with disabilities who had transport accessibility to health facilities had a twofold (AOR = 2.28; 95% CI [1.32–3.94]) higher likelihood of contraceptive use compared with those who had no transport accessibility. Regarding age, reproductive-age females with disabilities who were 25 to 34 years old had three (AOR = 3.04; 95% CI [1.53, 6.04) times higher odds of contraceptive use compared with those who were in the age group of 15 to 24 years old. Participants with hearing disabilities were 62% (AOR = 0.38; 95% CI [0.18–0.79]), those with extremity paralysis were 94% (AOR = 0.06; 95% CI 0.03–0.12]), and those with wheel-chair disabilities were 90% (AOR = 0.10; 95% CI [0.05–0.22]) less likely to use contraceptives than their counterparts with vision disabilities (Table 2).

## DISCUSSION

The prevalence of contraceptive method use among reproductive-age females with disabilities was 27.3%. In the multivariable multilevel logistic regression analysis, contraceptive knowledge, transport accessibility to the health facility, being an adult (25 to 34 years old), and types of disability were significantly associated with contraceptive use.

This study revealed that the contraceptive method use prevalence of 27.3% was almost similar to the studies conducted in Uganda, 26.1% (*Ayiga & Kigozi, 2016*), in Sierra Leone, 26.9% (*Trani et al., 2011*) and in Ethiopia, 24.5% (*Gonie et al., 2020*) and 27.2% (*Tsegay, Gebremariam & Haile, 2017*). On the other hand, the prevalence of contraceptive use in the current study is higher than in the previous studies conducted in Kenya, 16% (*Odhiambo, 2012*), Ghana, 17% (*Kumi-Kyereme, 2021*), and in Ethiopia 18% (*Beyene, Munea & Fekadu, 2019*). The possible reasons might be the long time interval between previous studies (2008) (*Odhiambo, 2012*) and our study (2022). Due to age group differences in the study population, school young people in the previous study (*Kumi-Kyereme, 2021*) and reproductive-age females with disabilities in the current study. Due to information bias from the source of data, since the information was collected from caregivers (*Odhiambo, 2012*). The other possible elucidation might be due to sample size differences. However, the prevalence of contraceptive use is lower than the studies conducted in Namibia, 32.7% (*Loeb & Grut, 2005*), Nigeria, 34% (*Olajide et al., 2014*) and in Ethiopia 34% (*Yesgat et al., 2020*). The possible justification might be that the Namibia study (*Loeb & Grut, 2005*) was conducted among married women with disabilities ,the chance of having unprotected

**Table 2  Multilevel logistic regression analysis for factors associated with contraceptive use among reproductive-aged females with disabilities in Sidama National Regional State, 2022.**

| Variables | | Contraceptive | | COR with 95% CI | AOR with 95% CI |
|---|---|---|---|---|---|
| | | Use | Not use | | |
| Marital status | Not married | 48 | 238 | 1.00 | 1.00 |
| | Married | 121 | 215 | 2.79 (1.92, 4.35)* | 1.53 (0.87, 2.69) |
| Education | Illiterate | 85 | 258 | 1.00 | 1.00 |
| | Literate | 84 | 198 | 1.30 (0.89, 1.91)* | 1.13 (0.68, 1.88) |
| Occupation | No occupation | 130 | 389 | 1.00 | 1.00 |
| | Have occupation | 39 | 62 | 1.88 (1.02, 2.68)* | 1.17 (0.63, 2.16) |
| Self-perception | Bad | 66 | 127 | 1.00 | 1.00 |
| | Good | 103 | 324 | 0.61 (0.41, 0.92)* | 0.99 (0.57, 1.70) |
| Age (Years) | 15 to 24 | 28 | 193 | 1.00 | 1.00 |
| | 25 to 34 | 88 | 153 | 3.96 (2.61, 7.22)* | 3.04 (1.53, 6.04)** |
| | 35 to 44 | 48 | 85 | 3.89 (1.89, 6.02)* | 2.20 (0.99, 4.85) |
| | 45 to 49 | 5 | 20 | 1.70 (0.52, 4.85) | 2.09 (0.56, 7.83) |
| Transport accessibility | Yes | 129 | 257 | 2.34 (1.52, 3.59)* | 2.28 (1.32, 3.94)** |
| | No | 40 | 194 | 1.00 | 1.00 |
| Distance to a health facility | ≥ 30 min on foot | 56 | 90 | 1.39 (0.84, 2.30)* | 1.17 (0.64, 2.11) |
| | <30 min on foot | 113 | 361 | 1.00 | 1.00 |
| Knowledge | Knowledgeable | 143 | 239 | 4.85 (2.97, 7.90)* | 9.03 (4.39, 18.6)** |
| | Not knowledgeable | 26 | 212 | 1.00 | 1.00 |
| Wealth index | Rich | 67 | 141 | 1.21 (0.76, 1.92) | 1.41 (0.77, 2.57) |
| | Medium | 46 | 150 | 0.99 (0.61, 1.63) | 1.22 (0.66, 2.24) |
| | Poor | 56 | 160 | 1.00 | 1.00 |
| Disability type | Hearing | 30 | 100 | 0.26 (0.14, 0.48)* | 0.38 (0.18, 0.79)** |
| | Extremity | 36 | 185 | 0.18 (0.10, 0.31)* | 0.06 (0.03, 0.12)** |
| | Wheel-chaired | 40 | 84 | 0.39 (0.22, 0.71)* | 0.10 (0.05, 0.22)** |
| | Vision | 63 | 82 | 1.00 | 1.00 |
| Residence | Urban | 50 | 189 | 0.65 (0.34, 1.21)* | 0.84 (0.42, 1.71) |
| | Rural | 119 | 262 | 1.00 | 1.00 |

**Notes.**
*P-value < 0.2.
**P-value < 0.05.
AOR, Adjusted odds ratio; CI, Confidence interval.

sex increased among married people, and people with disabilities have a higher desire to prevent pregnancy (*Casebolt et al., 2022*).

In this study, contraceptive knowledge is found to be significantly associated with contraceptive use. Those participants who had contraceptive knowledge had a higher chance of using contraceptives compared with those who had no contraceptive knowledge. The finding is consistent with the studies conducted in Uganda (*Ayiga & Kigozi, 2016*) and Nigeria (*Olajide et al., 2014*). The possible justification might be due to the power of knowledge to create awareness and overcome some cultural and social constraints that may act as a barrier to the use of contraceptives (*Beyene, Munea & Fekadu, 2019*).

Those who had transport availability to the health facility had a greater chance of contraceptive use when compared with those who had no transport accessibility to the health facility. As it is known, most people with disabilities face a physical challenge (*Olajide et al., 2014*) in accessing health facilities. Due to this, transportation is very important for accessing health facilities and getting contraceptive methods. Being an adult (25 to 34 years old) increased the chance of contraceptive use when compared with the age group of 15 to 24 years old. The possible justification is that the chance of marriage and unprotected sexual intercourse will increase among 25 to 34-year-old females with disabilities, and the chance of using contraception will also increase due to the greater desire of people with disabilities to avoid pregnancy (*Casebolt et al., 2022*). Compared with vision impairment, the probability of using contraceptives by hearing-disability reproductive age females with disabilities had 62% lower odds of contraceptive use, 94% lower odds of contraceptive use by extremity paralysis disabilities, and 90% lower odds by wheel-chaired disabilities. This finding is inconsistent with studies conducted in Gondar, the Amhara region, Ethiopia, and Addis Ababa. The studies revealed that the probability of contraceptive use increased among vision-impaired females with disabilities when compared with other types of disabilities (*Beyene, Munea & Fekadu, 2019*). The possible reason for the difference might be that visually impaired females with disabilities had an increased chance of accessing information access through different social media, most commonly radio. Radio is one of the most accessible and effective channels of information transmission for people with disabilities in developing countries, including Ethiopia (*Beyene, Munea & Fekadu, 2019*).

These findings may be important for different stakeholders who are concerned about reproductive-age females with disabilities and their reproductive health services, specifically contraceptive use. This study was conducted among all types of reproductive-age females with disabilities who reside in urban and rural areas. In the previous studies, rural residents with disabilities were excluded from the contraceptive use assessment studies. The other strength of this study was the use of multilevel analysis to check the effect of kebele-level variables on contraceptive use. However, due to the sensitivity and principles of contraceptive use, this study did not consider reproductive-age females with mental disabilities. Therefore, this study could be generalized to all reproductive-age females with disabilities except for mental disabilities.

## CONCLUSION

Contraceptive use among reproductive-age females with disabilities, specifically among the unmarried, is noticeably low in the Dale and Wonsho districts and Yirgalem city administration, Sidama National Regional State, Ethiopia. Contraceptive knowledge, accessible transportation to the health facility, being an adult (25 to 34 years old), and having specific types of disability were factors associated with contraceptive use among reproductive-age females with disabilities. Therefore, designing appropriate strategies to provide contraceptive education and information and provide contraceptive services in their homes is important to enhance contraceptive use.

## ACKNOWLEDGEMENTS

The authors would like to thank the data collectors and the study participants for their valuable contributions to this study. We are also grateful to the Sidama National Regional State Public Health Institute and selected woredas and kebeles for their assistance.

### Funding

The authors received no financial support for this study.

### Competing Interests

The authors declare there are no competing interests.

### Author Contributions

- Zelalem Tenaw conceived and designed the experiments, performed the experiments, analyzed the data, prepared figures and/or tables, authored or reviewed drafts of the article, data analysis, and approved the final draft.
- Taye Gari analyzed the data, prepared figures and/or tables, authored or reviewed drafts of the article, and approved the final draft.
- Achamyelesh Gebretsadik analyzed the data, prepared figures and/or tables, authored or reviewed drafts of the article, and approved the final draft.

### Ethics

The following information was supplied relating to ethical approvals (i.e., approving body and any reference numbers):

The Institutional Review Board at the College of Medicine and Health Sciences of Hawassa University approved the study (Ref.No: IRB/143/14.).

### Data Availability

The raw data is available in the Supplemental Files.

### Supplemental Information

Supplemental information for this article can be found online at http://dx.doi.org/10.7717/peerj.15354#supplemental-information.

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
