# Peer review of "Contraceptive use among reproductive-age females with disabilities in central Sidama National Regional State, Ethiopia: a multilevel analysis"

_PeerJ, doi:10.7717/peerj.15354_

## Round 0.1 · original submission · Major Revisions

The main argument of the manuscript is that the use of contraceptives among reproductive-age females with disabilities in central Sidama National Regional State, Ethiopia is a crucial issue that needs to be addressed. This study provides valuable insights into the determinants of contraceptive use among this population, utilizing a multilevel analysis approach. Overall, this is a well-conducted study that provides valuable insights into understanding the complex interplay between individual and contextual factors that influence contraceptive use among this population. A few minor revisions will further enhance the clarity and impact of the paper:

1) The introduction section could be further strengthened by highlighting the significance of the study area: Consider providing more context on the importance of studying contraceptive use among reproductive-age females with disabilities in Ethiopia. Clarify the research gap this study aims to fill.

2) The Methods section is well-detailed, but could benefit from a clearer explanation of how the Explain how the multilevel analysis was conducted and what it entailed. Consider discussing any limitations of the study design.

Reviewer 1 ·

Basic reporting

At the outset, I would like to congratulate the authors for conducting a research in the marginalised community at multiple domains-disability, gender and low income country, regarding an important issue in family welfare. The references are adequate and the reporting of the sections are appropriate and in correct order.
However, few comments and revisions are suggested to improve the manuscript
1. General comments:
The language of the manuscript can be improved further. Few sentences are not properly worded. Eg in Abstract “After adjusting for potential confounding variables, having good knowledge about contraceptives (AOR=9.03; 95% CI: 4.39, 18.6), transport accessibility to health facilities (AOR=2.28; 95% CI: 1.32, 3.94), being an adult (25 to 34 years old) (AOR=3.04; 95% CI: 1.53, 6.04), having a hearing disability (AOR = 0.38; 95% CI: 0.18, 0.79), paralysis of the extremities (AOR = 0.06; 95% CI: 0.03, 0.12) and wheel-chaired disability (AOR = 0.10; 95% CI: 0.05, 0.22) were factors associated with contraceptive use”

Experimental design

4. Methods and materials:
4.1 The research question, study design and settings have been described adequately
4.2 Sample Size: The following lines “The sample size for factors associated with contraceptive use was also computed using Epi-Info version 7 with the assumptions of a two-sided confidence level of 95%, a power of 80, a ratio of (unexposed: exposed), and a percent outcome in the unexposed group versus percent outcome in the exposed group. A” is not clear. What was the effect estimate (i.e odds ratio) assumed for the sample calculation?
4.3 Data collection procedures are explained adequately.
4.4 Statistical methods are applied appropriately
4.5 The study is ethically approved and the same has been conveyed by the authors

Validity of the findings

5. Results
5.1 The author says “Among the study participants, 382 (61.6%) had good knowledge about contraceptives”. How was the knowledge declared as “good”? any specific scoring or cut-offs?
5.2 Regarding the reasons for nor using contraceptives, the authors have not reported about the ones who are planning to have a baby in the near future in the text (although mentioned in the figure 1). This is an important factor, as it helps us to determine the unmet need of the contraceptives, and a few lines can be added in the discussion.
6. Discussion
6.1 Discussion is appropriate. Language of the discussion section needs improvement. Long sentences can be broken into simpler and smaller sentences, for the reader’ better understanding.
6.2 The authors say “The other possible justification might be due to sample size….”. Sample size per se does not influence the effect estimate, especially if it had been calculated with adequate power and errors. The author needs to clarify on the Nigeria study, if they did not calculate the sample size scientifically, before making this inference as a reason for the differential findings between the Nigeria study and present study

8. Conclusion:
The authors say “….arranging transportation (ambulance) is important to enhance contraceptive use”. If the authors are talking about the transport to ensure accessibility , why would an ambulance be required for transportation related to contraception? This can be sorted out by providing incentives/reimbursements for the women for using any existing public transport system to access the contraceptives, and if such transport systems are not available, suitable transport can be arranged by the health facilities for the disabled women. Alternatively, delivery of the contraceptives through the community health worker model can also be explored. It would be better if the authors discuss about these things.

Additional comments

2. Abstract:
The abstract is a concise version of the manuscript and the authors have done well to ensure that.

3. Introduction:
3.1 The author says “More than one billion people in the world are expected to have disabilities” Why would we expect them to have disabilities? Is the author meaning “estimated to have…”? Needs clarification.
3.2 The authors have made a very good case and rationale for their study in the Intriduction
3.3 The negative influence of disability on contraceptive use can be brought out by adding a few lines on the relationship between them in the introduction.

7. Conflict of interests, funding, data availability has been declared in the manuscript.

·

Basic reporting

No comment

Experimental design

Article is good in all the aspects of experimental design. Only thing lacking is about the procedure of selection of Kebeles in the methodology section.

Validity of the findings

No comment

Additional comments

If possible keep a table on contraceptive use prevalence with n, %, CI

---

## Round 0.2 · Minor Revisions

We have received a final review of your manuscript, the reviewer recommended revising the manuscript. We recommend rectifying grammatical errors, with thorough proofreading of your manuscript.

Reviewer 1 ·

Basic reporting

The authors have improved the language. Yet, there is scope for improvement of the language.

Experimental design

No comments

Validity of the findings

No comments

Additional comments

No comments

---

## Round 0.3 · accepted · Accept

The authors have addressed all of the reviewers' comments. The manuscript may be accepted as per the Journal Policy.